# Hot Cordilleran hinterland promoted lower crust mobility and decoupling of Laramide deformation

Dominik R. Vlaha [1,2,3] ✉, Andrew V. Zuza [1,3], Lin Chen [4] & Matthieu Harlaux[5]

The Late Cretaceous to Paleogene Laramide orogen in the North American Cordillera involved deformation >1,000 km from the plate margin that has been attributed to either plate-boundary end loading or basal traction exerted on the upper plate from the subducted Farallon flat slab. Prevailing tectonic models fail to explain the relative absence of Laramide-aged (ca. 90–60 Ma) contractional deformation within the Cordillera hinterland. Based on Raman spectroscopy of carbonaceous material thermometry and literature data from the restored upper 15–20 km of the Cordilleran crust we reconstruct the Late Cretaceous thermal architecture of the hinterland. Interpolation of compiled temperature data ($n = 200$) through a vertical crustal column reveals that the hinterland experienced a continuous but regionally elevated, upper-crustal geothermal gradient of >40 °C/km during Laramide orogenesis, consistent with peak metamorphic conditions and synchronous peraluminous granitic plutonism. The hot and partially melted hinterland promoted lower crust mobility and crust-mantle decoupling during flat-slab traction.

Intra-plate continental deformation remains inadequately quantified by plate tectonic theory because of the non-rigid behavior of the Earth's continents. Wide zones of intracontinental deformation result primarily from the weaker mineral phases that make up continental rocks and vertical strength heterogeneities and detachment levels in the crust compared to the more homogenous oceanic lithosphere[1,2]. Distributed intracontinental deformation occurs in regions of continental collision (i.e., the Himalaya-Tibetan orogen[3]), Andean-type oceanic subduction[4], and transform or divergent plate margins[5,6]. The style and distribution of such intra-plate deformation may be controlled by the thermal state and mechanical strength of the continental lithosphere[7–11]. In modern orogens, direct constraints on thermal architecture are limited to deep xenoliths and near-surface heat flow measurements, with geophysical measurements providing some indirect proxies[12–14]. Conversely, variably exhumed components of ancient orogens can provide insight into the thermal and structural architecture of intra-plate deformation[15].

The rheology and mechanical strength of the orogenic hinterland between the subduction-related magmatic arc and the retroarc fold-and-thrust belt impact the style, magnitude, dynamics, and extent of intracontinental deformation[7–11]. The physio-chemical properties within an orogenic hinterland govern lithospheric viscosity, which in turn impacts the intra-plate response to plate-boundary conditions[7,16]. Thus, the thermo-mechanical architecture of an exhumed hinterland can test leading geodynamic models for far-field deformation, including end loading[17] and basal traction[18]. We focus on the North American Cordillera hinterland region located between the Laramide uplifts and the Farallon-North American subduction boundary, which perplexingly experienced negligible Laramide-aged contractional deformation despite its central location[19].

The North American Cordillera experienced Middle Jurassic to early Cenozoic (ca. 170–50 Ma) contractional deformation during eastward subduction of the Farallon oceanic plate along the western margin of North America[20]. Protracted subduction of the Farallon plate

[1]Nevada Bureau of Mines and Geology, University of Nevada, Reno, NV, USA. [2]Department of Geological Sciences and Engineering, University of Nevada, Reno, NV, USA. [3]Nevada Geosciences, University of Nevada, Reno, NV, USA. [4]State Key Laboratory of Lithospheric Evolution, Institute of Geology and Geophysics, Chinese Academy of Sciences, Beijing, China. [5]BRGM - French Geological Survey, Orléans, France. ✉e-mail: dvlaha@unr.edu

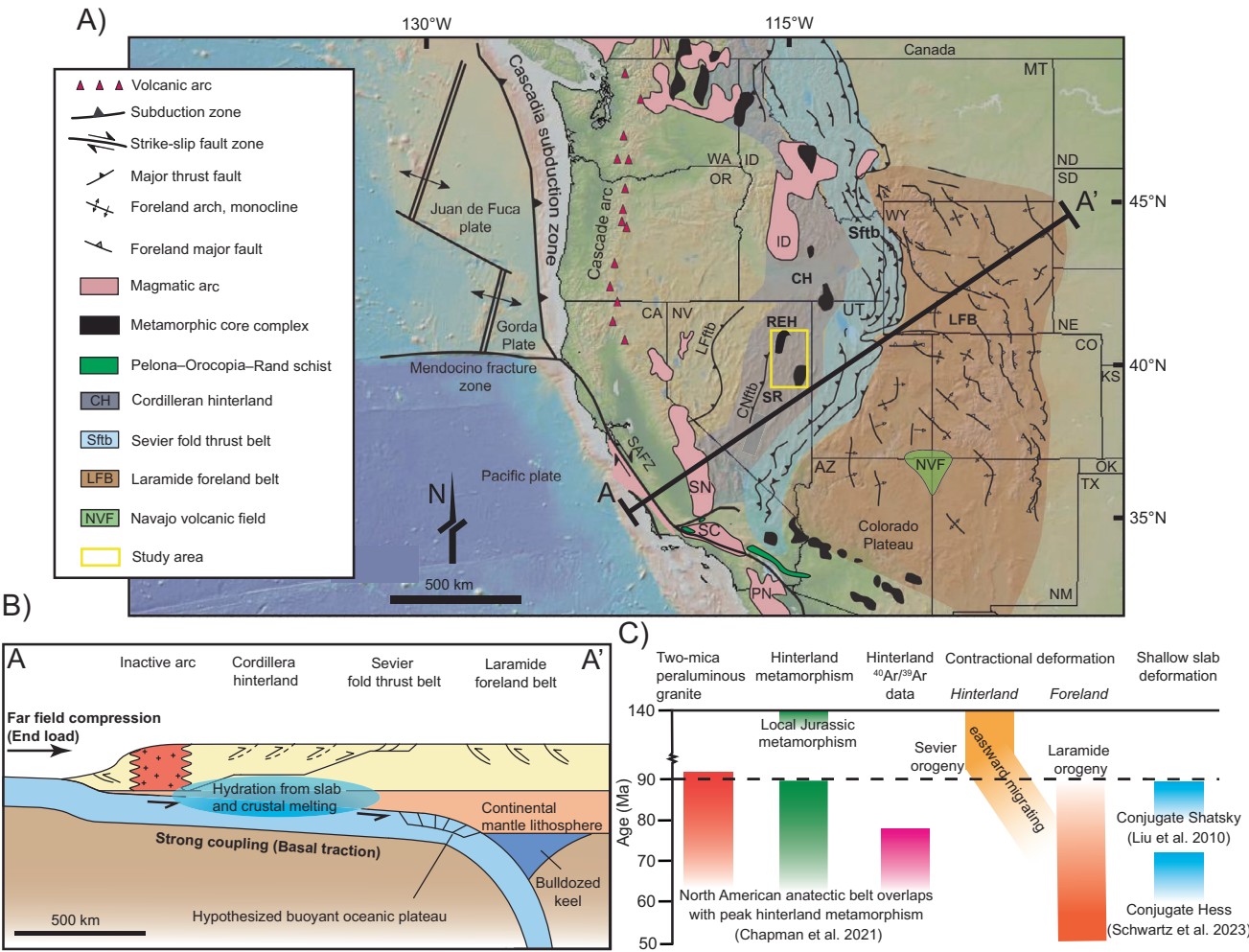

**Fig. 1 | Generalized map and schematic cross section of western North America showing competing models for Laramide deformation. A** Map of North America showing the location of major tectonic elements and the location of Fig. 2, modified from ref. 22. **B** Schematic cross section along A–A' showing competing models for Laramide tectonism (end loading vs. basal traction). Dashed faults within the Cordillera hinterland are expected for both models, however, there are no observed Laramide-aged contractional structures. **C** Temporal evolution of two-mica peraluminous granite generation, metamorphism, argon thermochronology data, contractional deformation, and flat slab subduction. Abbreviations for states:

AZ Arizona, CA California, CO Colorado, ID Idaho, KS Kansas, MT Montana, NE Nebraska, ND North Dakota, NM New Mexico, NV Nevada, OK Oklahoma, OR Oregon, SD South Dakota, UT Utah, WA Washington, WY Wyoming. Abbreviations for geologic elements: CNftb central Nevada fold-thrust belt, ID Idaho batholith, LFftb Luning–Fencemaker fold-and-thrust belt, PN Peninsular complex, REH Ruby Mountains–East Humboldt Range metamorphic core complex, SAFZ San Andreas fault zone, SC Southern California batholith, SN Sierra Nevada batholith, SR Snake Range metamorphic core complex.

generated two phases of deformation that partially overlap in space and time: (i) the Early to Late Cretaceous Sevier fold-and-thrust belt (ca. 125–66 Ma), characterized by thin-skinned deformation[21,22], and (ii) the Late Cretaceous to Paleogene Laramide orogeny (ca. 90–50 Ma), characterized by thick-skinned basement-cored uplifts and associated basins[22,23]. The majority of Sevier deformation occurred before the onset of Laramide orogenesis ca. 90 Ma, followed by a diminished eastward migrating trend of thrusting toward the foreland (Fig. 1C)[20,22]. The Laramide orogeny is particularly enigmatic because it consisted of a wide (>700 km) zone of basement-cored reverse faults located >1000 km inboard from the continental margin[20]. It is generally hypothesized that Laramide orogenesis was driven by flat-slab subduction of the northeast-dipping Farallon slab[24], which may have transmitted stresses to induce upper-crust strains via end loading[17] or basal traction from the subhorizontal oceanic slab[18,23]. Postulated flat-slab subduction is primarily evidenced by seismic tomography interpretations and plate reconstructions[25], hydrated and sheared mantle xenoliths[26,27], underplated Pelona-Orocopia-Rand (POR) schists in the

southwest United States[28], and the time transgressive sweep of Laramide tectonism and associated magmatism[18] (Fig. 1). Conversely, hit-and-run models predict oblique collision of the Insular superterrane with western North America at ca. 100–85 Ma, supported by paleomagnetic data, to drive Laramide deformation thus obviating the need for flat-slab subduction[29,30]. These models all explicitly or implicitly favor a mechanically rigid continental crust to transmit far-field stress to the continental interior.

The Cordilleran hinterland broadly records Middle Jurassic to Cretaceous plutonism, metamorphism, and distributed low-magnitude shortening[19,31–35], which was followed by a widespread phase of Late Cretaceous to early Cenozoic (ca. 90–60 Ma) regional metamorphism, crustal anatexis with peraluminous plutonism originating from partial melting of pelites and orthogneisses, syn-convergent extension, and negligible contractional deformation[15,32,33,36–41]. Published thermochronology and pressure–temperature–time (P-T-t) data from mid-crustal rocks exhumed in the Ruby Mountains–East Humboldt Range (REH) and

Snake Range metamorphic core complexes (Fig. 1) show that peak metamorphic conditions were attained in the Late Cretaceous to early Cenozoic (Supplementary File), contemporaneous with the postulated shallowing of the Farallon plate and start of Laramide orogeny within the continental interior. Late Cretaceous peak metamorphic conditions were only minorly and locally overprinted by subsequent slab rollback magmatism and metamorphic core complex exhumation[15]. Therefore, most rocks along the margins of metamorphic core complexes experienced peak temperatures coeval with peak metamorphic conditions in the Late Cretaceous.

Here, we present structural and thermal reconstructions from the hinterland of the North American Cordillera to provide an unprecedented picture of the thermal state and mechanical strength of intraplate deformation during the Late Cretaceous–early Cenozoic Laramide orogeny (Fig. 1). This contribution determines the Late Cretaceous peak temperature conditions experienced by the upper 15–20 km of crust via Raman spectroscopy of carbonaceous material (RSCM) thermometry to constrain the bulk thermal state and rheology of the mid-upper crust. Our results show that the Cordillera hinterland crust experienced elevated thermal conditions during Laramide orogenesis, which would have promoted lower crustal melting and mechanical decoupling from the subducting Farallon flat slab. These findings provide thermal constraints for the hinterland region, thus giving insights into the geodynamic processes that drive far-field intra-plate deformation.

## Results

We generated a large dataset of new and compiled peak temperatures ($n = 200$) from along the margins of the REH and Snake Range metamorphic core complexes (Supplementary Dataset 1 and 2). This dataset mostly consists of temperatures derived from RSCM thermometry ($n = 145$) and lesser Conodont alteration index (CAI) ($n = 33$), calcite–dolomite thermometry ($n = 12$), garnet–biotite thermometry and garnet–muscovite–biotite–plagioclase thermobarometry (GARB/GMBP) ($n = 7$), and titanium-in-quartz thermometry (TitaniQ) ($n = 3$) data. We restored the structural position of the temperature estimates to reconstruct their initial depth position in the Late Cretaceous, prior to regional Cenozoic extension (Supplementary Fig. 1). This allows for robust establishment of thermal profiles across the Cordilleran hinterland. Due to the similarity and overlap in temperature versus depth trends, we merged the REH and Snake Range data into a composite dataset that is considered as representative of the hinterland (Fig. 2B).

To establish the ambient crustal geothermal gradient, we isolated the lowest-quartile (coldest) samples at a given depth interval ("Methods"). This is because (i) the dominant thermometry method (RSCM) records peak thermal conditions, and (ii) there are multiple processes that could result in anomalous, local sample heating (e.g., hydrothermal fluids, local intrusions, shear-induced heating and faulting), but none that can uniquely cool a sample. The lowest-quartile temperature versus binned depth intervals shows a monotonic thermal gradient with increasing temperatures at depth across the upper ~20 km of crust (Fig. 2C). Compiled argon thermochronology and P-T-t data indicate that these temperatures were attained and persisted throughout the Late Cretaceous and early Cenozoic (ca. 90–60 Ma) (Fig. 2A and Supplementary File).

Monte Carlo simulations were used to construct steady-state conductive thermal profiles that best fit our observed temperature dataset with reasonable thermal parameters ("Methods"), which allowed extrapolation of the upper-middle crust data to Moho depths and temperatures. Moho temperatures typically vary from 700–900 °C in most non-arc settings[14,42,43]. Our thermal models were guided by a Moho temperature of ~800 °C at the base of thickened orogenic plateau crust (i.e., 60-km thick[20,22]).

The best-fit thermal results match the temperature in the restored upper ~15–20 km of the Late Cretaceous crust of the Cordilleran hinterland. Specifically, the regional geothermal gradient in the upper 13 km is elevated at 45.9 °C/km ($R^2 = 0.978$), that transitions to a cooler slope with depth (Fig. 2C). This observed thermal structure is comparable to modern continental backarcs, including the Central Andes[42].

## Discussion

The well-constrained crustal temperature profile can be used to estimate crustal strength and viscosity over a range of compositions (Fig. 3)[44]. Hot crustal temperatures imply a reduced viscosity across the entire crust, but the most significant viscosity reduction would occur due to partial melting when temperatures exceed the solidus at depths >30 km (Fig. 3). Such melting would be concentrated within the middle to lower crust to reduce the bulk strength and dynamic viscosity by over two orders of magnitude (Fig. 3; "Methods"). The feldspathic or diopside-rich phases in the mid-lower crust[44] may inhibit melting relative to the chosen wet-granite solidus, but focused melting of specific felsic granite or pelitic layers could yield significant bulk-viscosity reductions throughout the integrated mid-lower crust[45]. This is consistent with partial melting of metapelites and orthogneisses to generate the voluminous Late Cretaceous to early Paleogene (ca. 90–40 Ma) peraluminous plutons observed throughout the Cordilleran hinterland[36]. For comparison, we show strength and viscosity curves calculated for a cooler typical continental geotherm of ~24 °C/km in the upper crust (surface heat flow of ~65 mW m⁻²) (Fig. 3)[46,47]. The Laramide foreland likely had a cooler upper-crustal paleo-geothermal gradient of ~20 °C/km[22]. Most geodynamic models assume a comparably cool thermal structure for the hinterland region[17,18,48] that is inconsistent with our temperature estimates.

The distribution of Late Cretaceous two-mica peraluminous granites across the Cordillera and similar along-strike metamorphic cooling ages suggest that the mid-to-lower crust must have been weak, hot, and mobile beneath significant extents of the Cordilleran hinterland[36]. The exact mechanism for this heating and melting is debated but was likely caused by the combined processes of crustal thickening, radiogenic heating, and slab-related metamorphic devolatilization[15,36,49]. Partial melting of the mantle lithospheric wedge and lower crust would advect peraluminous magmas and heat to the mid-crust[15,36,49]. Regardless of the origin of regional crustal heating, the timing of peak metamorphic conditions and melting in the hinterland is synchronous with postulated Laramide deformation to the east (Figs. 1C and 4 and Supplementary File).

Our dataset provides unique temperature estimates to test disputed models for Laramide deformation, including (i) plate-boundary driven models including end-loading during flat slab subduction[17] or hit-and-run terrane accretion models[29] versus (ii) flat-slab traction models[18,23]. An applied end load along the plate margin is associated with the collision of the conjugate Shatsky Rise[17,25], conjugate Hess Rise[50], or Insular superterrane[29] that would have resulted in a compressional stress state across the North American lithosphere to induce far-field deformation. Conversely, basal traction models predict that shear stress is transmitted from the subducting low-angle slab to the base of the overriding continental crust, where deformation broadly follows the trajectory of the subducted oceanic plateau[18,22,23]. However, geodynamic models that recreate both end-member scenarios, which have also been applied to modern orogens (e.g., Sierras Pampeanas), invoke a mechanically rigid continental crust where deformation would only occur at pre-existing structural or rheological heterogeneities[17,18,48].

The observed hot and weak hinterland crust is inconsistent with end-loading or hit-and-run models because the mechanically weak crust could not transmit stresses from the plate boundary to the Laramide foreland belt[42,51] without inducing observable Laramide-aged contractional deformation across the hinterland. Instead, the reconstructed thermal structure of the hinterland crust supports that basal

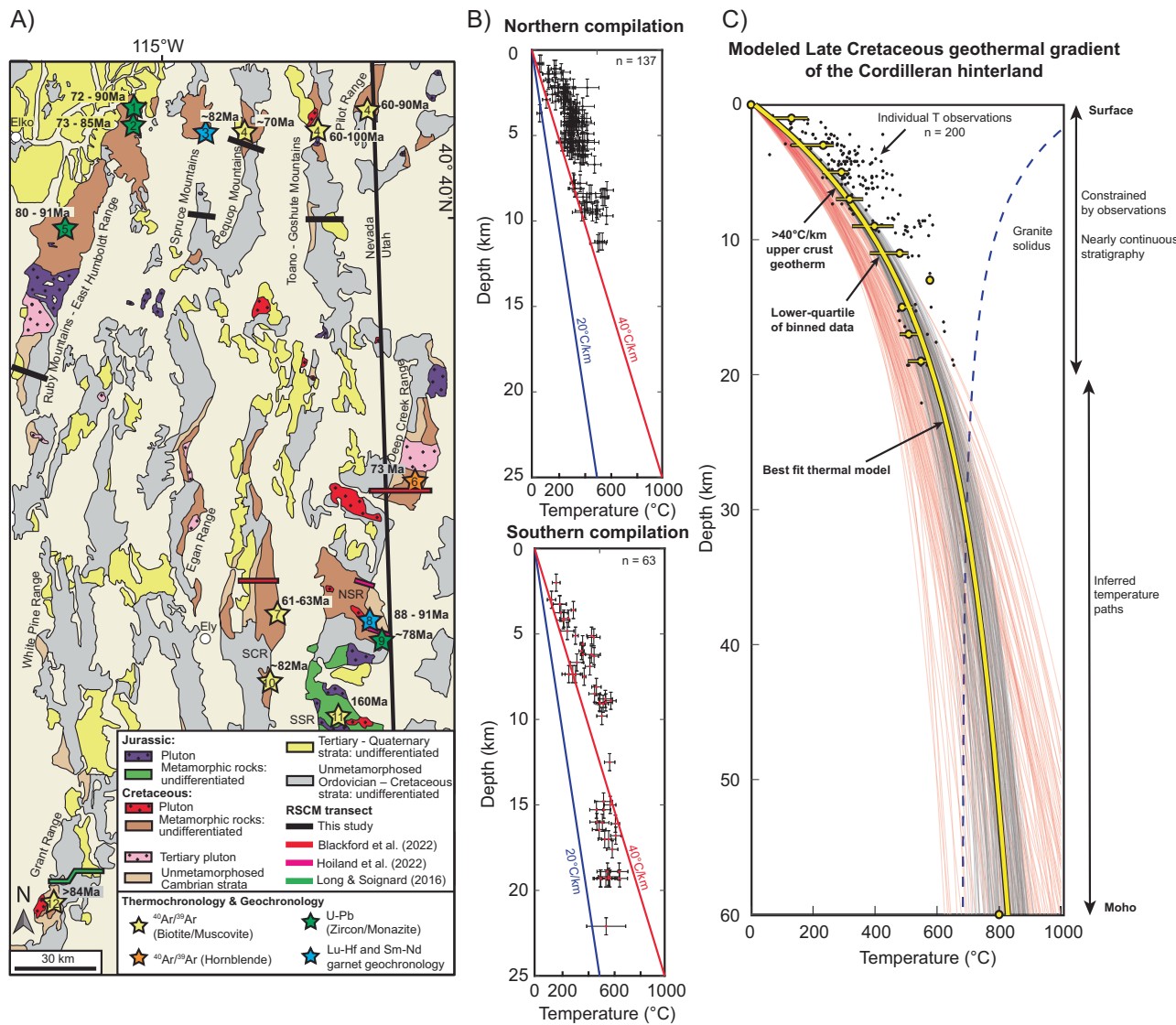

**Fig. 2 | Thermal modeling results. A** Geologic map showing locations of Jurassic and Cretaceous plutons and metamorphism in the Cordillera hinterland[15,72]. RSCM transects are highlighted in black, red, pink, and green from this study and refs. [72–75], respectively. Stars indicate the locations and ages of thermochronology and geochronology data; for additional sources, see Supplementary File. NSR northern Snake Range, SCR Schell Creek Range, SSR southern Snake Range. **B** Temperature versus depth plots from the northern and southern thermal datasets. Data is plotted against linear contoured thermal gradients. Temperature is reported with 2 standard errors after ref. [63], while depth is reported with ±0.5 km uncertainty. **C** Modeled Late Cretaceous geotherm of the Cordilleran hinterland. The best fit thermal model is highlighted in yellow and is plotted with best-fit thermal profiles (black) and acceptable thermal profiles (red) from our Monte Carlo simulation ("Methods").

traction from the subducting flat slab induced Laramide tectonism (Fig. 4). Basal traction imparted on the upper-plate continental mantle lithosphere (CML) may have bulldozed or translated parts of the lithosphere inboard[17,52], but such strains would have been decoupled from the upper crust by the melt-rich low-viscosity middle-to-lower crust in the hinterland[23,42]. The top of the low-angle Farallon slab was shallow in the west (~35 km depth) as evidenced by the underplated POR schists[28], but this geometry transitioned to a more moderate ~120 km depth in the east as evidenced by eclogite xenoliths[26] in the Navajo volcanic field (Fig. 1). This depth transition implies a hinge where the shallow slab encountered stronger CML as it translated inboard[52]. Therefore, there would have been enhanced coupling of the Farallon slab with the colder, stronger foreland and the bulldozed keel[17,51,53–55] (Fig. 4 and Supplementary File), consistent with high differential stress observations from mantle xenoliths obtained from the Navajo volcanic field[27] (Fig. 1). The absence of a compressional stress state in the previously thickened hinterland crust may have allowed an

early phase of syn-convergent extension within the weak hinterland crust[41].

An alternative interpretation is that the mechanically strong upper CML drove Laramide foreland deformation via end loading despite the thermally weakened continental crust. Our data does not constrain the rigidity of the upper CML. However, we favor basal shear models to explain the more-widespread phase of basement-cored thrusting for the following reasons. If a mechanically rigid CML translated eastward to drive Laramide deformation, we would expect some upper-crust contractional strain in the weak hinterland crust, which is not observed. Furthermore, available data suggest that the CML did not have a high viscosity in the Late Cretaceous, including evidence of hydrated and sheared mantle xenoliths[26,27] and the interpreted eastward translation of continental arc lower-crust (arclogite) beneath the Colorado plateau[56]. Hydration-induced weakening of the CML during flat-slab subduction explains the observed CML deformation. Lastly, Laramide tectonism and associated magmatism

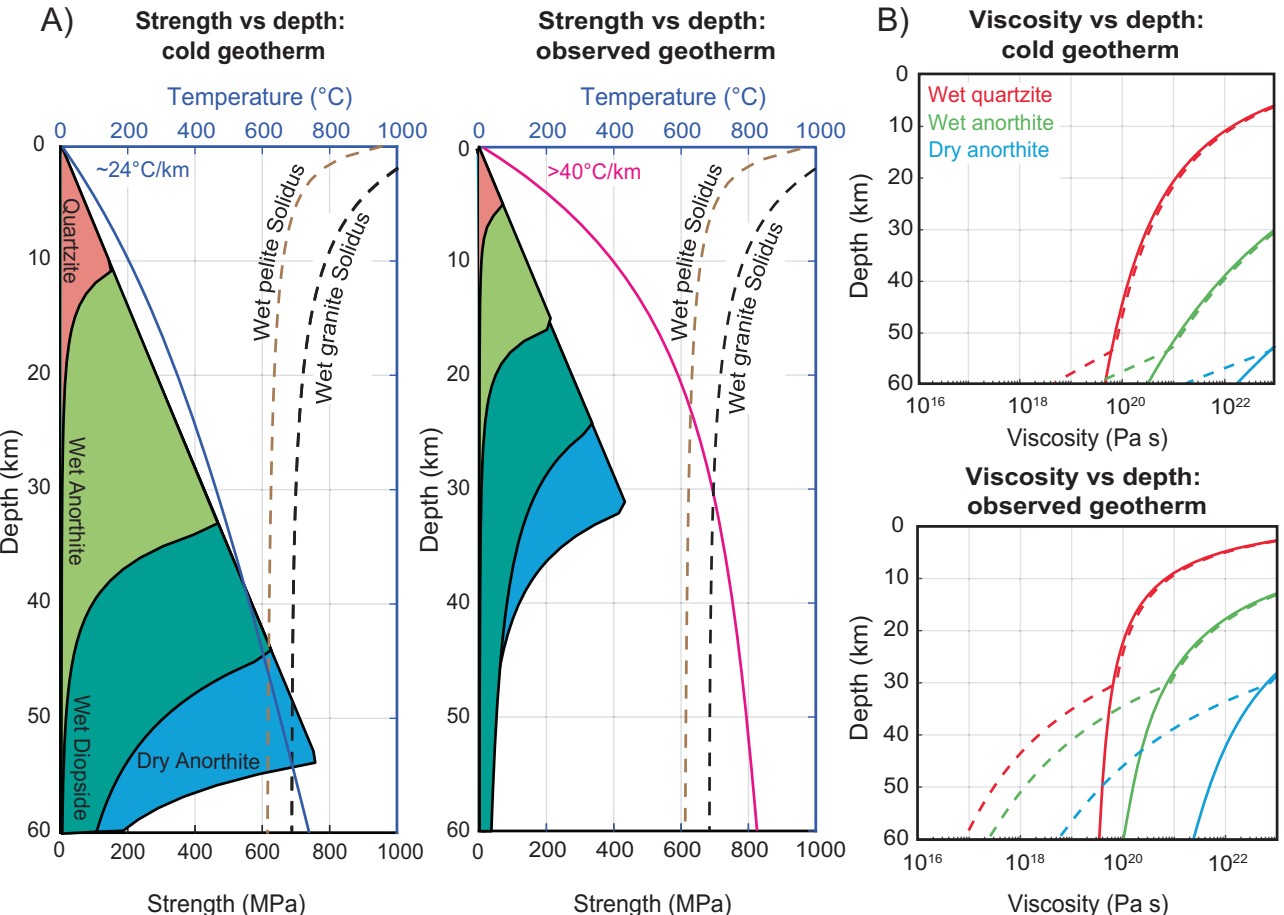

**Fig. 3 | Crustal strength and viscosity profiles for the Late Cretaceous Cordilleran hinterland versus a typical continental geotherm. A** Modeled continental yield stress profiles based on rheology (wet quartzite[66]; wet and dry anorthite[67]; wet diopside[68]) and our observed thermal structure ("Methods"). Our results are compared to a cold, typical geothermal gradient using values derived from refs. 46,47. **B** Modeled viscosity versus depth profiles comparing the two thermal structures. Solid and dashed lines indicate the effective viscosity without and with partial melt generation, respectively ("Methods").

displays a time-transgressive northeastward sweep[18], which is not explained by end-loading.

Our interpretations from the Laramide hinterland are similar to geophysical observations from other modern flat-slab locations. For example, magnetotelluric and receiver function studies from the subhorizontal Cocos plate in central Mexico show an enhanced conductivity and low-velocity anomaly zone interpreted as partial melting and/or dehydrating fluids in the mid-crust[57,58]. The analogous melt-rich low-velocity zone is interpreted to promote strain decoupling, explaining the lack of compressional structures in the overriding plate since the Miocene[58].

The onset timing of postulated flat-slab subduction is critical to relating this process to crustal heating, the generation of peraluminous granites[36], and Laramide orogenesis[25,50]. Flat-slab subduction has been attributed to either underthrusting of the buoyant oceanic conjugate Shatsky plateau (ca. 90–80 Ma)[25] or the conjugate Hess plateau (ca. 75 Ma)[50]. The presence of underplated trench-derived schists in southern California and Arizona is commonly cited as evidence for shallow slab subduction[28]. Maximum depositional ages from detrital zircon datasets derived from the schist exposures display a northwest (ca. 95–80 Ma) to southeast (75–60 Ma) younging distribution[28]. However, active arc magmatism in the southern California batholith (Fig. 1) as young as ca. 75 Ma suggests that shallow slab subduction did not occur before this time, challenging the need for underthrusting of the conjugate Shatskey plateau[50].

Late Cretaceous Farallon subduction rates of 10–12 cm/year[22] suggest that potential slab-related influence on upper-plate

peraluminous magmatism and Laramide orogenesis may have been delayed by ca. 10 Myr after the initial subduction of the oceanic plateau. For example, an early phase of subduction at ca. 90–80 Ma predicts slab-related devolatilization and crustal melting would advect heat through the continental crust by ca. 70 Ma, explaining the observed elevated thermal state and peak metamorphism in the hinterland[15]. Alternatively, ca. 75 Ma subduction of the conjugate Hess plateau[50] implies that ca. 90–60 Ma crustal heating and melting must have been initially driven by another process[36], with potential late-stage influence from Farallon-slab hydration of the continental lithosphere. However, subduction of the conjugate Hess plateau is consistent with the general timing of major thick-skin basement-cored thrusting, increased exhumation, and basin development in Colorado, Utah, and Wyoming ca. 70–50 Ma[18,50,52,59,60].

Previous Laramide models assume a homogenous thermomechanical structure in the Cordillera hinterland region[17,18] whereas our results demonstrate a relatively hot and weak hinterland that would not permit transmission of end-loaded stresses to the Laramide interior. Our working hypothesis is that the subducting flat slab hydrated the mantle lithosphere and lower crust to drive partial melting[49,57] that advected heat to the mid-upper crust, which is consistent with the spatiotemporal onset of flat-slab subduction, peraluminous granite plutonism across the hinterland, and peak regional metamorphic conditions (Fig. 4). This would have resulted in a competition between melt-induced heat advection and refrigeration from the colder oceanic slab. However, our observation of an elevated thermal state in the upper-mid crust (Fig. 2C) requires that heating

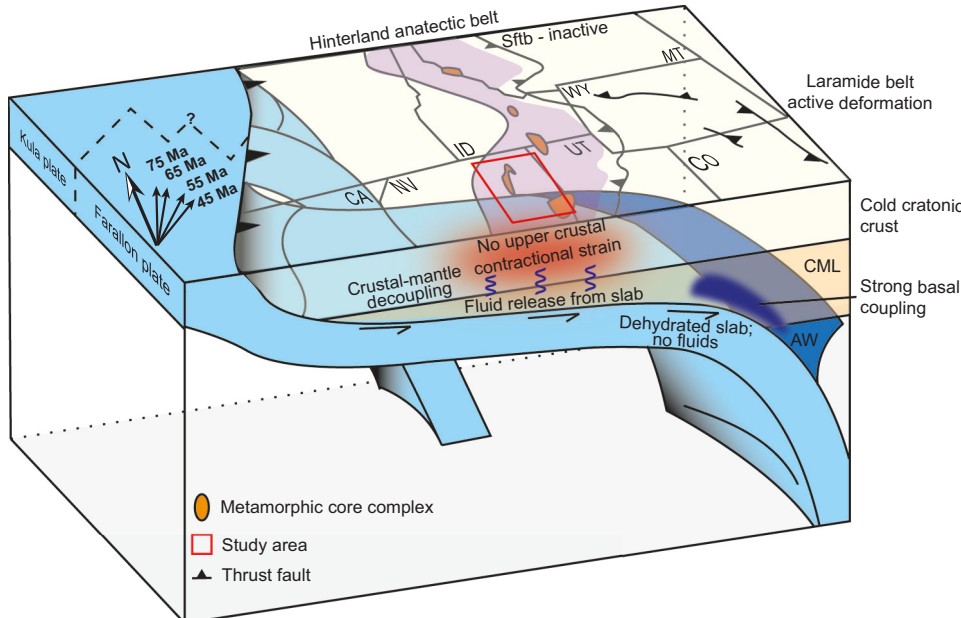

**Fig. 4 | Modified basal traction model.** Schematic block diagram palinspastically restored to the Late Cretaceous-Paleogene after ref. 17 showing the inferred thermal and mechanical state during Laramide flat-slab subduction. Black arrows in the upper left indicate approximate relative convergence direction between Farallon and North America plates at times noted[18]. Dashed line represents the approximate Farallon-Kula ridge[76]. AW asthenospheric wedge, CML continental mantle lithosphere, Sftb Sevier fold-and-thrust belt. State abbreviations are consistent with Fig. 1.

dominated for most crustal levels, potentially due to a deep slab or delayed conductive cooling[55]. In the colder foreland, slab refrigeration of the CML would enhance coupling, allowing basal shear to transmit through the crust (Fig. 4). Future geodynamic models must consider the elevated thermal state and weakened mechanical strength of the hinterland to test tectonic models that explain far-field intra-plate deformation (e.g., basal traction[18]; end loading[17]; hit-and-run[29,30]).

## Methods

### Timing and paleodepth of peak thermal conditions
Compiled pressure temperature and time (P-T-t) data and argon thermochronology indicate regional temperatures were attained in the Late Cretaceous and persisted to the Paleocene (ca. 90–60 Ma). Here, we compile published geochronology and thermochronology to estimate the relative age of RSCM peak temperature estimates. Compiled geochronology consists of U-Pb dating of metamorphic zircon, monazite, and titanite, and $^{176}Lu/^{177}Hf$ and $^{147}Sm/^{144}Nd$ garnet geochronology (Fig. 2; Supplementary File). Geochronology results constrain the timing of peak metamorphic conditions, whereas argon thermochronology tracks cooling through a specific closure temperature for biotite, muscovite, hornblende, and K-feldspar, and a combination of different thermochronometers within the sample can record a time-transgressive cooling history.

Pre-extensional paleodepth estimates in this study were drafted and restored slip along major Cenozoic structures across five mountain ranges around the margins of the REH. Cross sections were constrained via geologic mapping and published geological maps (Supplementary File). Basin and Range extensional rotation and paleosurface is recorded by a regional Paleogene subvolcanic unconformity (Supplementary File).

In summary, available evidence constrains that the highest observed temperatures in eastern Nevada were attained in the Late Cretaceous to early Cenozoic, ca. 90–60 Ma (Supplementary File). Later heating was less significant and did not impact the peak temperature results from the RSCM dataset. Paleodepths were constrained by pre-Cenozoic palinspastic reconstructions (Supplementary Fig. 1) to create compiled temperature versus depth datasets.

### Raman spectroscopy of carbonaceous material thermometry
Raman spectroscopy analyses were performed at the University of Nevada, Reno using a Horiba LabRAM HR Evolution spectrometer optimized for 200–2200 nm and equipped with an open space confocal microscope with 5 objectives (×5, ×10, ×50, ×50 LWD, and ×100), a Marzhauser XY motorized stage, two diffraction gratings (600 and 1800 gr/mm), and a multichannel CCD detector for a wide spectral resolution range. The excitation beam is provided by a frequency doubled Nd:YAG laser (Oxxius, France) at 532 nm with a maximum power of 100 mW and a beam diameter of 1 μm. Acquisition parameters were as follows: 600–2000 $cm^{-1}$ range, neutral-density filter of 1–10%, and 200 μm confocal hole. The signal-to-noise ratio was optimized, ideally lower than 1%, by adjusting the acquisition time and accumulation number. The variability of each sample was tested by analyzing 15 individual grains of carbonaceous material (CM). Our sampling strategy focused on dark organic-rich shale, limestone, and conglomerate. The laser was focused on CM located beneath a transparent grain (typically quartz or calcite), based on procedures defined in[61,62]. Peak fitting and temperature estimates are conducted by IFORS, an automated software package which processes iterative, random-based curve fitting using pseudo-Voigt functions[63,64]. From the package, we analyzed reference samples from ref. 63 to quantify a calibration curve derived from our spectrometer. Average temperature estimates are recorded from the area ratio and shape of the disordered carbon (D) and graphite (G) peaks[63,65]. All well-defined CM spectra produce D-band peaks near 1350 $cm^{-1}$ and a G-band peak near 1580 $cm^{-1}$[65]. Raw IFORS data are reported in Supplementary Dataset 2.

### Steady-state geothermal gradient
Monte Carlo simulations were used to construct a reasonable steady-state geotherm that fit the compiled temperature dataset. To calculate the steady-state geotherm, we follow the classic derivation from ref. 47 that considers exponential decay of radiogenic heating within the crust:

$$q_s = q_m + \rho h_r H_0 \qquad (1)$$

$$T(y) = T_0 + \frac{q_m y}{k} + \frac{(q_s - q_m)h_r}{k}\left(1 - e^{-y/h_r}\right) \qquad (2)$$

where $T_0$ is the temperature at depth $y = 0$, $q_s$ is the surface heat flux, $q_m$ is the mantle heat flux, $k$ is the thermal conductivity, $h_r$ is the length scale for the decrease of mean radioactive heat generation per unit mass, $\rho$ is density, and $H_0$ is the radiogenic heat production. Values were randomly selected within a range of reasonable values[46,47] in our temperature simulation (Supplementary File, Table S1). We computed one million iterations and best fit thermal profiles were calculated via normalized root mean square error (NRMSE), a statistical method that compares modeled data to our observed temperatures. Best-fit thermal profiles have an NRMSE value <0.25, while acceptable curves are <0.5 (Fig. 2C). Our best-fit thermal model is the mean of all best-fit curves, which is used for our rheology and viscosity calculations.

## Rheology and viscosity profiles

We construct first order strength envelopes of the continental lithosphere modeling wet quartzite[66], wet and dry anorthite[67], and wet diopside[68] for the upper and lower crust, respectively. For brittle frictional sliding, we follow Byerlee's Law[69]:

$$\tau = \mu_s(\sigma_n - P_f) \qquad (3)$$

where $\tau$ and $\sigma_n$ are shear and normal stress, respectively, $P_f$ is pore-fluid pressure, and $\mu_s$ is the coefficient of sliding friction, where $\mu_s$ equals 0.85 when $\sigma_n < 200$ Mpa for most rock types. For the ductile regime, rheology and viscosity calculations are modeled after ref. 44. Following theoretical and experimental rock deformation investigations showing that strain rate and stress fit the dislocation creep power law:

$$\dot{\varepsilon} = A f^r_{H_2O} \sigma^n \exp\left(-\frac{Q + PV}{RT}\right) \qquad (4)$$

where $\dot{\varepsilon}$ is the strain rate, $A$ is a material constant, $f_{H_2O}$ is water fugacity (see Eq. 2 in ref. 44), $r$ is the fugacity exponent, $\sigma$ is stress, $n$ is the stress exponent, $Q$ is the activation energy, $P$ is the pressure, $V$ is the activation volume, $R$ is the gas constant, and $T$ is temperature in Kelvin. Assuming a constant strain rate of $10^{-14}\,\mathrm{s}^{-1}$, viscosity $\eta$ is derived from the constitutive equation:

$$\eta = \frac{\sigma}{2\dot{\varepsilon}} \qquad (5)$$

Additionally, our modeled viscosity profiles account for partial melting using experimentally obtained P-T-dependent wet solidus and dry liquidus curves[70]. It is assumed that volumetric melt fraction $M$ increases linearly between the solidus and liquidus temperatures at a given pressure[70]:

$$M = \begin{cases} 0 & \text{when } T \leq T_{solidus} \\ \frac{T - T_{solidus}}{T_{liquidus} - T_{solidus}} & \text{when } T_{solidus} < T < T_{liquidus} \\ 1 & \text{when } T \geq T_{liquidus} \end{cases} \qquad (6)$$

where $T_{solidus}$ and $T_{liquidus}$ are the solidus and liquidus temperatures at a given depth, respectively. When the melt fraction exceeds 0.5% ($M_0$), the melt-weakening effect on viscosity ($\eta$) starts to work such that the effective viscosity ($\eta_{eff}$)[71]:

$$\eta_{eff} = \eta \exp(-28(M - M_0)). \qquad (7)$$

## Data availability
The RSCM and compiled temperature versus depth data generated in this study are provided in the Supplementary Information, Supplementary Dataset 1–3, and Figshare (https://doi.org/10.6084/m9.figshare.25555245).

## Code availability
This manuscript uses IFORS to process RSCM data[63,64]. IFORS can be downloaded here: http://www.sediment.uni-goettingen.de/download/.

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

## Acknowledgements
We thank Charles Thorman and Drew Levy for providing additional samples to analyze in the Pequop mountains, Spruce mountains, and the Toano range. This work was supported by the U.S. Geological Survey National Cooperative Geologic Mapping Program (STATEMAP and EDMAP) G14AC00237 (A.V.Z), G16AC00186 (A.V.Z), G17AC00212 (A.V.Z), G18AC00198 (A.V.Z), G19AC00383G (A.V.Z), G20AS00006 (A.V.Z), G21AC10873 (A.V.Z), G21AS00005 (A.V.Z), GS22AS00006 (A.V.Z), National Science Foundation Tectonics Program grant EAR 2210074 (A.V.Z), and student research grants from the Geological Society of America and Nevada Petroleum and Geothermal Society (D.R.V.).

## Author contributions
D.R.V. and A.V.Z. designed research; D.R.V. and A.V.Z. conducted fieldwork and sampling; D.R.V. and M.H. did laboratory analyses; D.R.V., A.V.Z. and L.C. did tectonic interpretations; all authors contributed to writing and editing.

## Competing interests
The authors declare no competing interests.
