## [Peer Review File · Nature Communications]

REVIEWER COMMENTS

Reviewer #1 (Remarks to the Author):

Review of Vlaha et al. "Hot Cordilleran hinterland promoted lower crustal mobility and decoupling of Laramide deformation"

Vlaha et al. report new and compiled thermal data from the western North American Cordillera hinterland with the goal of understanding processes that controlled Laramide orogenesis. The data are high quality and the modeling is excellent. The authors convincingly show that the Cordillera hinterland was hot and weak during this critical time period. This suggestion is not exactly new as others have shown that the hinterland was partially molten (i.e., 'anatectic belt': Chapman et al. 2021) and underwent protracted metamorphism (Wells et al., 2012). What is new about this article is the link to big picture tectonic events, specifically how strain was partitioned in the orogen in the Late Cretaceous. The new data and their creative interpretation are exciting and worth publishing in Nature Communications. The figures are also excellent and illustrate the data and the model very well. I predict that this article will be well received and widely discussed.

Having recently written an article on this same topic (Laramide orogeny), I'm particularly interested in timing implications for the data presented in this manuscript. I find the argument against end loading quite convincing, but my main criticism of the manuscript right now is that I'm not entirely clear on *when* basal traction occurred in their model and whether this process can explain both the spatial extent and duration of the Laramide orogeny. These parts of the manuscript could be developed more to both strengthen the article and tie the hinterland data more directly to the Laramide uplifts. For example, here's my concern: if basal traction due to subduction of a shallow slab is driving contractional deformation in the Laramide belt (rather than end-loading), there must be a significant timing delay between when the slab flattens at the trench and when contractional deformation occurs far inboard in the Laramide belt. Given Farallon spreading rates (10-14 cm/year), there could be a delay of 10+ myr from the onset of shallow-slab subduction until the development of Laramide-style basement uplifts. There are good constraints on when shallow slab processes started from shut down of the arc and underthrusting of trench sediments in Southern California (i.e., the location of proposed shallow-slab subduction) and these data can be used to calculate the timing of basal traction-driven deformation in the Laramide belt (see Schwartz et al., 2023 for timing details). From my calculations, basal traction driven contraction in the Laramide belt could not have begun earlier than 65 Ma based on data presented in Schwartz et al. (2023). This has implications for the development of Laramide basins and uplifts. I suggest that the authors develop their model further and place some timing constraints on the process that they propose based on their data and other constraints from the geologic record.

Another more minor point is that the role of extension in the hinterland is not discussed, but could very much support the model of a weak and collapsing(?) hinterland decoupled from the subducting slab. Throughout the manuscript the authors equate 'deformation' with 'contraction' (e.g., lines 57 and 61), but it is well documented that syn-convergent extension was occurring in the Late Cretaceous in the hinterland (see Wells and Hoisch, 2008; Wells et al., 2012). In figure 4, the authors show 'no upper crustal strain' which is not correct in detail. I think the authors are thinking of 'contractional strain' but

they should be aware of and discuss the Wells et al. (2012) model for extension and decoupling (see for example their figure 9). The model presented in this manuscript is quite different and uses a different dataset to arrive at the conclusion that the crust and mantle were decoupled; nonetheless, they should incorporate the structural data for extension into their model.

Overall, this is an excellent manuscript and I like it a lot. The new data, modeling and interpretation will be a fantastic contribution to the ongoing discussion of the causes of the Laramide orogeny. I recommend the paper be considered for publication with Nature Communications after moderate revision. Feel free to contact me if anything is unclear. Best of luck with the revisions.

Specific comments:

Line 57: There is good evidence for deformation, just not contraction. There is lots of evidence of extension. Change 'deformation' to 'contraction'.

Line 61: 'deformation' to 'contraction'

Lines 68-76: The role of Late Cretaceous extension in the hinterland is missing here and is an important part of the story. See Wells and Hoisch (2008) and Wells et al. (2012).

Line 125: typo: 'were guided by a were guided by a'

Line 161: the reference to conjugate Hess Rise should be Schwartz et al. (2023).

References:

Comment: Some references are incomplete and some lack proper capitalization. See some highlights below:

Reference #3: Capitalize 'Science'

Reference #8: Capitalize 'Geoscience'

Reference #26: This has no journal or book information. It appears to be a GSA special paper

Reference #31: Incomplete capitalization of title

Reference #44: See comment above. I didn't see it referenced in the manuscript but should be on Line 161? Also need to capitalize 'Communications'

Reference #65: This has no journal or book information. It appears to be a book chapter in a GSA special

Figures:

Figure 2: The inset map was hard to follow. For example, the grey field is labeled 'Mountain range polygon.' What does that mean? What rocks are they? Under Jurassic and Cretaceous there are boxes for 'metamorphism.' Do you mean 'metamorphic rocks undifferentiated'?

Figure 4: delete 'no upper crustal strain' and consider changing to 'extensional faulting'

Reviewer #2 (Remarks to the Author):

This manuscript addresses a debated issue: what causes intra-plate deformation that far from the convergent margin. There are two classic proposed mechanisms, end loading and basal traction, which could far-field thick-skinned deformation. The author focused on the typical far-field thick-skinned deformation, the Laramide Orogeny in the Late Cretaceous. They utilized Raman spectroscopy on carbonaceous material thermometry to restore the thermal state during the Laramide deformation. Their results show an elevated thermal condition during the Laramide time, and they believe that the higher thermal state could lead to lower crustal melting and decoupling from the Farallon flat slab. The authors argue that a hot and weak hinterland inhabits stress transferring to continental interior. As a result, they prefer the basal shear model than end-loading model.

I read through the manuscript with interest and pleasure. Note that I cannot judge the details of methods about thermochronology, but based on other studies using same methods, they provided trustworthy results. Overall, I think that this paper would make an interesting contribution to the understanding of intra-plate large-scale deformation. However, before acceptance for publication, it needs to address several issues and other minor modifications. I suggest a minor-moderate revision. The main issue is that the authors suggest that far-field deformation is more likely due to basal shear than end loading because a hot and weak hinterland inhibits stress transfer from trench to continental interior. However, the upper-most of continental mantle lithosphere can be strong (higher viscosity), might be able to transmit stress laterally. I understand the authors' samples do not have upper mantle material, but a discussion is needed. Another discussion needed is there is layer of weak mantle lithosphere in between the flat slab and lower crust as the Farallon flat slab is at a depth >120km (e.g., Usui et al., 2003; Hernández-Urib and Palin, 2019; Liu and Currie, 2019). The mantle lithosphere right above the flat slab could have low viscosity which cannot transmit the shear traction upward. I suggest adding more discussion on the two mechanisms (end-loading and basal shear).

Below are some minor suggestions:

L20-21: "subducting flat oceanic slab" change to "subducted Farallon flat slab".

L45: "heatflow" to "heat flow".

L56-L57: I suggest referring it as the Farallon-North American subduction boundary because this might be confused with the westward subduction idea (which is east of the hinterland region; English and Johnston, 2004; Johnston and Borel, 2007; Johnston, 2008; Hildebrand, 2009)

L65: "decoupling from the subhorizontal Farallon slab". In my opinion, this phrase makes it sound like the flat slab is right beneath continental crust. Maybe rephrase?

L68: Before this paragraph, it was a nice introduction. Maybe add a new section here name "tectonic setting"?

L77: "The Laramide orogeny in western North America....>1,000 km inboard from the continental margin", add reference.

L92: "Ruby Mountains-East Humboldt Range (REH)..." add reference to figure 1.

L125: Remove “were guided by a”. It appeared twice in the sentence.

L126: what about a thinner crust?

L153-L156: “The exact mechanism for this heating...plutons and heat to the mid-crust”. I Suggest divide into two sentences. Hard to read.

L169: “end-member models”. You mean end-loading models? The current phase indicates both end-loading and basal shear, which is contradict with your argument in the following paragraph.

L172: “weak crust could not transmit stresses”. What about upper most of continental mantle lithosphere? The top layer of mantle lithosphere is strong, could this layer transmit stress? I recommend to add discussion about this.

L178-L182: The Farallon flat slab depth is predicted to be >120 km depth and even deeper beneath the Laramide. However, mantle lithosphere right above the flat slab has low viscosity. How basal shear transmitted upward if there is a weak layer existed between crust and the flat slab?

L257: “table” to “Table”

Figures:

Figure 1: I suggest adding a few more ticks for latitude and longitude.

Figure 3: The plots titles do not have the same fonts.

**REVIEWER COMMENTS**

**Reviewer #1**

Review of Vlaha et al. "Hot Cordilleran hinterland promoted lower crustal mobility and
decoupling of Laramide deformation"

Vlaha et al. report new and compiled thermal data from the western North American Cordillera
hinterland with the goal of understanding processes that controlled Laramide orogenesis. The
data are high quality and the modeling is excellent. The authors convincingly show that the
Cordillera hinterland was hot and weak during this critical time period. This suggestion is not
exactly new as others have shown that the hinterland was partially molten (i.e., 'anatectic belt':
Chapman et al. 2021) and underwent protracted metamorphism (Wells et al., 2012). What is
new about this article is the link to big picture tectonic events, specifically how strain was
partitioned in the orogen in the Late Cretaceous. The new data and their creative interpretation
are exciting and worth publishing in Nature Communications. The figures are also excellent and
illustrate the data and the model very well. I predict that this article will be well received and
widely discussed.

Thank you very much for the kind remarks. We appreciate your constructive review, below.

Having recently written an article on this same topic (Laramide orogeny), I'm particularly
interested in timing implications for the data presented in this manuscript. I find the argument
against end loading quite convincing, but my main criticism of the manuscript right now is that
I'm not entirely clear on *when* basal traction occurred in their model and whether this process
can explain both the spatial extent and duration of the Laramide orogeny. These parts of the
manuscript could be developed more to both strengthen the article and tie the hinterland data
more directly to the Laramide uplifts. For example, here's my concern: if basal traction due to
subduction of a shallow slab is driving contractional deformation in the Laramide belt (rather
than end-loading), there must be a significant timing delay between when the slab flattens at the
trench and when contractional deformation occurs far inboard in the Laramide belt. Given
Farallon spreading rates (10-14 cm/year), there could be a delay of 10+ myr from the onset of
shallow-slab subduction until the development of Laramide-style basement uplifts. There are
good constraints on when shallow slab processes started from shut down of the arc and
underthrusting of trench sediments in Southern California (i.e., the location of proposed shallow-
slab subduction) and these data can be used to calculate the timing of basal traction-driven
deformation in the Laramide belt (see Schwartz et al., 2023 for timing details). From my
calculations, basal traction driven contraction in the Laramide belt could not have begun earlier
than 65 Ma based on data presented in Schwartz et al. (2023). This has implications for the
development of Laramide basins and uplifts. I suggest that the authors develop their model
further and place some timing constraints on the process that they propose based on their data
and other constraints from the geologic record.

We found this to be a very insightful comment and we agree. The timing of postulated basal
shear is critical and its consideration provides quantitative, testable estimates to build our
preferred tectonic model. We have incorporated this analysis in the main text, and copy our new
text below. Thank you for this suggestion.

The onset timing of postulated flat-slab subduction is critical to relating this
process to crustal heating, the generation of peraluminous granites³⁶, and Laramide
orogenesis^{25,50}. Flat-slab subduction has been attributed to either underthrusting of the

buoyant oceanic conjugate Shatsky plateau (ca. 90–80 Ma)²⁵ or the conjugate Hess
plateau (ca. 75 Ma)⁵⁰. The presence of underplated trench-derived schists in southern
California and Arizona is commonly cited as evidence for shallow slab subduction²⁷.
Maximum depositional ages from detrital zircon datasets derived from the schist
exposures display a northwest (ca. 95–80 Ma) to southeast (75–60 Ma) younging
distribution²⁸. However, active arc magmatism in the southern California batholith as
young as ca. 75 Ma suggests that shallow slab subduction did not occur before this time,
challenging the need for underthrusting of the conjugate Shatskey plateau⁵⁰.

Late Cretaceous Farallon subduction rates of 10–12 cm/yr²² suggest that potential
slab-related influence on upper-plate peraluminous magmatism and Laramide
orogenesis may have been delayed by ca. 10 Myr after the initial subduction of the
oceanic plateau. For example, an early phase of subduction at ca. 90–80 Ma predicts
slab-related devolatilization and crustal melting would advect heat through the
continental crust by ca. 70 Ma, explaining the observed elevated thermal state and peak
metamorphism in the hinterland¹⁵. Alternatively, ca. 75 Ma subduction of the conjugate
Hess plateau⁵⁰ implies that ca. 90–60 Ma crustal heating and melting must have been
initially driven by another process³⁶, with potential late-stage influence from Farallon-
slab hydration of the continental lithosphere. However, subduction of the conjugate Hess
plateau is consistent with the general timing of major thick-skin basement-cored
thrusting, increased exhumation, and basin development in Colorado, Utah, and
Wyoming ca. 70–50 Ma^{19,50,52,59,60}.

Another more minor point is that the role of extension in the hinterland is not discussed, but
could very much support the model of a weak and collapsing(?) hinterland decoupled from the
subducting slab. Throughout the manuscript the authors equate ‘deformation’ with ‘contraction’
(e.g., lines 57 and 61), but it is well documented that syn-convergent extension was occurring in
the Late Cretaceous in the hinterland (see Wells and Hoisch, 2008; Wells et al., 2012). In figure
4, the authors show ‘no upper crustal strain’ which is not correct in detail. I think the authors are
thinking of ‘contractional strain’ but they should be aware of and discuss the Wells et al. (2012)
model for extension and decoupling (see for example their figure 9). The model presented in
this manuscript is quite different and uses a different dataset to arrive at the conclusion that the
crust and mantle were decoupled; nonetheless, they should incorporate the structural data for
extension into their model.

This comment is greatly appreciated. We have included the role of syn-convergence extension
in both the section on tectonic setting and discussion. This does not directly impact or modify
our model, but provides an external observation that the onset of interpreted brittle extension in
the Late Cretaceous hinterland region may reflect a mid-lower crust that was mobile and
decoupling from the underlying mantle, for example as discussed by Wells et al. (2012
Tectonics). Please see the modified text, and the newly incorporated text copied below:

**The absence of a compressional stress state in the previously thickened hinterland crust
may have allowed an early phase of syn-convergent extension within the weak hinterland
crust⁴¹. (Wells et al., 2012).**

Overall, this is an excellent manuscript and I like it a lot. The new data, modeling and
interpretation will be a fantastic contribution to the ongoing discussion of the causes of the
Laramide orogeny. I recommend the paper be considered for publication with Nature
Communications after moderate revision. Feel free to contact me if anything is unclear. Best of
luck with the revisions.

104

Thank you for taking the time to review our manuscript. We appreciate your helpful
critiques which have significantly improved the quality of this manuscript.

Specific comments:

Line 57: There is good evidence for deformation, just not contraction. There is lots of evidence
of extension. Change 'deformation' to 'contraction'.

This is correct. We improved the clarity by specifying contractional deformation throughout the
manuscript.

Line 61: 'deformation' to 'contraction'
We have made this modification. Thank you.

Lines 68-76: The role of Late Cretaceous extension in the hinterland is missing here and is an
important part of the story. See Wells and Hoisch (2008) and Wells et al. (2012).
You are correct, thank you for this comment. As discussed above, we now have included syn-
convergent extension in this section of the manuscript. We also include the role of decreased
compressional stress and increased gravitational potential energy in our discussion.

Line 125: typo: 'were guided by a were guided by a'
Thank you for catching this typo. We have corrected it.

Line 161: the reference to conjugate Hess Rise should be Schwartz et al. (2023).
Yes, that is the correct reference. Thank you for catching this typographic mistake.

References:

Comment: Some references are incomplete and some lack proper capitalization. See some
highlights below:

Reference #3: Capitalize 'Science'

Reference #8: Capitalize 'Geoscience'

Reference #26: This has no journal or book information. It appears to be a GSA special paper

Reference #31: Incomplete capitalization of title

Reference #44: See comment above. I didn't see it referenced in the manuscript but should be
on Line 161? Also need to capitalize 'Communications'

Reference #65: This has no journal or book information. It appears to be a book chapter in a
GSA special

Thank you for these edits. We have carefully gone through our references to make sure they
are formatted and displayed properly.

Figures:

Figure 2: The inset map was hard to follow. For example, the grey field is labeled 'Mountain
range polygon.' What does that mean? What rocks are they? Under Jurassic and Cretaceous
there are boxes for 'metamorphism.' Do you mean 'metamorphic rocks undifferentiated'?

You are correct. We have increased the size of the inset key to include these specifications. We
have also changed 'metamorphism' in the legend to 'metamorphic rocks undifferentiated'.

Lastly, the gray polygons are labeled 'Unmetamorphosed Ordovician – Cretaceous strata
undifferentiated'.

Figure 4: delete 'no upper crustal strain' and consider changing to 'extensional faulting'

Thank you for the comment, we have changed the text to: **No upper crustal contractional**
**strain.**

**Reviewer #2**

This manuscript addresses a debated issue: what causes intra-plate deformation that far from
the convergent margin. There are two classic proposed mechanisms, end loading and basal
traction, which could far-field thick-skinned deformation. The author focused on the typical far-
field thick-skinned deformation, the Laramide Orogeny in the Late Cretaceous. They utilized
Raman spectroscopy on carbonaceous material thermometry to restore the thermal state during
the Laramide deformation. Their results show an elevated thermal condition during the
Laramide time, and they believe that the higher thermal state could lead to lower crustal melting
and decoupling from the Farallon flat slab. The authors argue that a hot and weak hinterland
inhabits stress transferring to continental interior. As a result, they prefer the basal shear model
than end-loading model.

I read through the manuscript with interest and pleasure. Note that I cannot judge the details of
methods about thermochronology, but based on other studies using same methods, they
provided trustworthy results. Overall, I think that this paper would make an interesting
contribution to the understanding of intra-plate large-scale deformation. However, before
acceptance for publication, it needs to address several issues and other minor modifications. I
suggest a minor-moderate revision.

Thank you for taking the time to review our manuscript. We appreciate your helpful constructive
comments, which we have carefully considered and agree with. Based on these suggestions,
we have made significant changes and improvements to the manuscript. Thank you again.

The main issue is that the authors suggest that far-field deformation is more likely due to basal
shear than end loading because a hot and weak hinterland inhibits stress transfer from trench to
continental interior. However, the upper-most of continental mantle lithosphere can be strong
(higher viscosity), might be able to transmit stress laterally. I understand the authors' samples
do not have upper mantle material, but a discussion is needed. Another discussion needed is
there is layer of weak mantle lithosphere in between the flat slab and lower crust as the Farallon
flat slab is at a depth >120km (e.g., Usui et al., 2003; Hernández-Urib and Palin, 2019; Liu
and Currie, 2019). The mantle lithosphere right above the flat slab could have low viscosity
which cannot transmit the shear traction upward. I suggest adding more discussion on the two
mechanisms (end-loading and basal shear).

You are absolutely correct. Considering the strength of continental mantle lithosphere (CML)
was thought provoking, and additional work must be conducted on the dynamics between the
bulldozed keel and the CML. We have added a new discussion addressing the timing
constraints and the two mechanisms (end-loading and basal shear). Please see the modified
text, and the newly incorporated text copied below:

Basal traction imparted on the upper-plate continental mantle lithosphere (CML)
may have bulldozed or translated parts of the lithosphere inboard^{18,52}, but such strains
would have been decoupled from the upper crust by the melt-rich low-viscosity middle-
to-lower crust in the hinterland^{23,42}. The top of the low-angle Farallon slab was shallow in
the west (~35 km depth) as evidenced by the underplated POR schists²⁸, but this
geometry transitioned to a more moderate ~120 km depth in the east as evidenced by
eclogite xenoliths²⁶ in the Navajo volcanic field (Fig. 1). This depth transition implies a
hinge where the shallow slab encountered stronger CML as it translated inboard⁵².
Therefore, there would have been enhanced coupling of the Farallon slab with the colder,
stronger foreland and the bulldozed keel^{18,51,53–55} (Fig. 4; Supplementary file), consistent
with high differential stress observations from mantle xenoliths obtained from the Navajo
volcanic field²⁷ (Fig. 1). The absence of a compressional stress state in the previously
thickened hinterland crust may have allowed an early phase of syn-convergent extension
within the weak hinterland crust⁴¹.

An alternative interpretation is that mechanically strong upper CML drove
Laramide foreland deformation via end loading despite the thermally weakened
continental crust. Our data does not constrain the rigidity of the upper CML. However, we
favor basal shear models to explain the more-widespread phase of basement-cored
thrusting for the following reasons. If a mechanically rigid CML translated eastward to
drive Laramide deformation, we would expect some upper-crust contractional strain in
the weak hinterland crust, which is not observed. Furthermore, available data suggest
that the CML did not have a high viscosity in the Late Cretaceous, including evidence of
hydrated and sheared mantle xenoliths^{26,27} and the interpreted eastward translation of
continental arc lower-crust (“arclogite”) beneath the Colorado plateau⁵⁶. Hydration-
induced weakening of the CML during flat-slab subduction explains the observed CML
deformation. Lastly, Laramide tectonism and associated magmatism displays a time-
transgressive northeastward sweep¹⁹, which is not explained by end-loading.

Below are some minor suggestions:

L20-21: “subducting flat oceanic slab” change to “subducted Farallon flat slab”.

Thank you, we made this change.

L45: “heatflow” to “heat flow”.

Thank you for catching this error. This has been corrected throughout the manuscript and
supplemental file.

L56-L57: I suggest referring it as the Farallon-North American subduction boundary because this
might be confused with the westward subduction idea (which is east of the hinterland region;
English and Johnston, 2004; Johnston and Borel, 2007; Johnston, 2008; Hildebrand, 2009)

Thank you for helping us clarify this statement. We have read through the references and agree,
we do not want people to be confused with the westward subduction idea. We have made
changes to the text.

L65: “decoupling from the subhorizontal Farallon slab”. In my opinion, this phrase makes it sound
like the flat slab is right beneath continental crust. Maybe rephrase?

I agree, this statement makes it seem like the flat slab is right beneath the continental crust. We
changed the sentence to match our edits from L81-L82.

L68: Before this paragraph, it was a nice introduction. Maybe add a new section here name
“tectonic setting”?

You are correct, this improves the organization of the manuscript.

L77: “The Laramide orogeny in western North America....>1,000 km inboard from the
continental margin”, add reference.

We added a reference, thank you.

L92: “Ruby Mountains-East Humboldt Range (REH)...” add reference to figure 1.

We added the reference to figure 1, thank you.

L125: Remove “were guided by a”. It appeared twice in the sentence.

Thank you for catching this typo, it is greatly appreciated.

L126: what about a thinner crust?

This is a great point, and a question I also considered when conducting these models. Changing
the depth of the crust does not significantly impact the overall thermal structure. I tested these
data for a 40-km thick crust and the solidus depth decreases by a few kilometers. This results in
a slightly hotter thermal structure and does not impact the upper crustal geothermal gradient.

L153-L156: “The exact mechanism for this heating...plutons and heat to the mid-crust”. I
Suggest divide into two sentences. Hard to read.

Great suggestion and I agree. We divided this sentence.

L169: “end-member models”. You mean end-loading models? The current phase indicates both
end-loading and basal shear, which is contradict with your argument in the following paragraph.
You are correct, this sentence alone contradicts the following paragraph. I have added a phrase
that clarifies our thoughts. Here, we are referring to the broad geodynamic models which have
been applied to modern orogens (e.g., Sierras Pampeanas). The following paragraph is now
discussing a modified version of the basal shear model.

L172: “weak crust could not transmit stresses”. What about upper most of continental mantle
lithosphere? The top layer of mantle lithosphere is strong, could this layer transmit stress? I
recommend to add discussion about this.

You are correct, thank you for this thought-provoking comment. Please see the modified text,
and the newly incorporated text copied below. This paragraph has been used in a previous
comment in the text above:

**An alternative interpretation is that mechanically strong upper CML drove**
**Laramide foreland deformation via end loading despite the thermally weakened**
**continental crust. Our data does not constrain the rigidity of the upper CML. However, we**

favor basal shear models to explain the more-widespread phase of basement-cored
thrusting for the following reasons. If a mechanically rigid CML translated eastward to
drive Laramide deformation, we would expect some upper-crust contractional strain in
the weak hinterland crust, which is not observed. Furthermore, available data suggest
that the CML did not have a high viscosity in the Late Cretaceous, including evidence of
hydrated and sheared mantle xenoliths^{26,27} and the interpreted eastward translation of
continental arc lower-crust (“arclogite”) beneath the Colorado plateau⁵⁶. Hydration-
induced weakening of the CML during flat-slab subduction explains the observed CML
deformation. Lastly, Laramide tectonism and associated magmatism displays a time-
transgressive northeastward sweep¹⁹, which is not explained by end-loading.

L178-L182: The Farallon flat slab depth is predicted to be >120 km depth and even deeper
beneath the Laramide. However, mantle lithosphere right above the flat slab has low viscosity.
How basal shear transmitted upward if there is a weak layer existed between crust and the flat
slab?

This is an excellent point. Thank you for bringing this to our attention. Overall, we argue that
initial slab refrigeration of the CML enhances coupling and allows stress to transfer through the
crust. Please see the modified text, and the newly incorporated text copied below. The first
paragraph has been used in a previous comment in the text above:

**Basal traction imparted on the upper-plate continental mantle lithosphere (CML)**
**may have bulldozed or translated parts of the lithosphere inboard^{18,52}, but such strains**
**would have been decoupled from the upper crust by the melt-rich low-viscosity middle-**
**to-lower crust in the hinterland^{23,42}. The top of the low-angle Farallon slab was shallow in**
**the west (~35 km depth) as evidenced by the underplated POR schists²⁸, but this**
**geometry transitioned to a more moderate ~120 km depth in the east as evidenced by**
**eclogite xenoliths²⁶ in the Navajo volcanic field (Fig. 1). This depth transition implies a**
**hinge where the shallow slab encountered stronger CML as it translated inboard⁵².**
**Therefore, there would have been enhanced coupling of the Farallon slab with the colder,**
**stronger foreland and the bulldozed keel^{18,51,53–55} (Fig. 4; Supplementary file), consistent**
**with high differential stress observations from mantle xenoliths obtained from the Navajo**
**volcanic field²⁷ (Fig. 1). The absence of a compressional stress state in the previously**
**thickened hinterland crust may have allowed an early phase of syn-convergent extension**
**within the weak hinterland crust⁴¹.**

This would have resulted in a competition between melt-induced heat advection and
refrigeration from the colder oceanic slab, but our observation of an elevated thermal state in
the upper-mid crust (Fig. 2C) requires that heating dominated for most crustal levels, potentially
due to a deep slab or delayed cooling⁵⁵. **In the colder foreland, slab refrigeration of the CML**
**would enhance coupling, allowing basal shear to transmit through the crust (Fig. 4).**

L257: “table” to “Table”

Thank you for catching this. This was corrected.

Figures:

Figure 1: I suggest adding a few more ticks for latitude and longitude.

We added a few more graticule ticks for latitude and longitude.

Figure 3: The plots titles do not have the same fonts.

Thank you for pointing this out. We have checked the fonts and made sure they are the same
for all titles in the figures.

REVIEWERS' COMMENTS

Reviewer #1 (Remarks to the Author):

Review of Vlaho et al. "Hot Cordilleran hinterland promoted lower crustal mobility and decoupling of Laramide deformation"

This is an excellent manuscript and the authors have addressed all of my major concerns in this draft. I have only a few minor editorial points below. Therefore, I enthusiastically recommend the paper for publication with Nature Communications. The article will make a very nice contribution to the Laramide orogeny controversy!

Specific comments:

Lines 24-25: vary word choice: you use 'reconstructed' and 'reconstruct' in same sentence.

Line 28: hyphenate 'upper-crustal' since it modifies geothermal gradient

Line 39: 'weaker' than what? What minerals specifically?

Lines 92-93: In this sentence, it sounds like you are say that end-loading, basal traction, and Insular superterrane collision 'all explicitly or implicitly favor a mechanically rigid continental crust to transmit far-field stresses.' Is this what you mean to say?

Line 94: no hyphen after 'Middle'

Figures:

Figure 1: The Southern California Batholith is left off the Cordilleran arc in western North America. It was a significant, 500 km-long belt of arc magma that was in 'flare-up' mode during the beginning of the Laramide orogeny. I would add a pink field for the Mojave and Transverse ranges and label it as the 'SC' or 'SCB' to denote that there was magmatism there in the Late Cretaceous. This is of course an important part of the Laramide story...

Figure 4: The figure is fine, but you should be aware that the Farallon-Kula ridge was present after 85 Ma. I'm not sure that you want to put it in this schematic figure, but depending on the timing, the Farallon plate label may be on the 'Kula plate'. See Umhoefer (1987): Umhoefer, P.J., 1987. Northward translation of "BAJA British Columbia" along the Late Cretaceous to Paleocene margin of western North America. *Tectonics*, 6(4), pp.377-394.

Reviewer #2 (Remarks to the Author):

I have thoroughly reviewed the revised version of the manuscript "Hot Cordilleran hinterland promoted lower crust mobility and decoupling of Laramide deformation", submitted by Vlaho et al. Following my previous comments and suggestions, I have evaluated the authors' revisions and the response letter that accompanied the submission.

I am pleased to report that the authors have addressed all the concerns raised during the initial review.

The revisions made to the manuscript, including adding extra discussion about the two mechanisms, end-loading and basal shear, and the additional discussion regarding the other reviewer's comments on the timing constraints on the Laramide events have significantly improved the quality and clarity of the work. The authors have provided comprehensive responses to the feedback, and the modifications made are well-justified and effectively incorporated into the manuscript.

Based on the thoroughness of the revisions and the quality of the manuscript in its current form, I recommend the manuscript for publication in Nature communication

**REVIEWER COMMENTS**

**Reviewer #1**

Review of Vlaha et al. "Hot Cordilleran hinterland promoted lower crustal mobility and
decoupling of Laramide deformation"

This is an excellent manuscript and the authors have addressed all of my major concerns in this
draft. I have only a few minor editorial points below. Therefore, I enthusiastically recommend the
paper for publication with Nature Communications. The article will make a very nice contribution
to the Laramide orogeny controversy!

Thank you for taking the time to review our manuscript. We appreciate your helpful
critiques which have significantly improved the quality of this manuscript and the contribution to
the Laramide orogeny controversy. This comment is greatly appreciated.

Specific comments:

Lines 24-25: vary word choice: you use 'reconstructed' and 'reconstruct' in same sentence.
We have made this modification to the abstract, thank you.

Line 28: hyphenate 'upper-crustal' since it modifies geothermal gradient

Thank you, we made this change.

Line 39: 'weaker' than what? What minerals specifically?

Thank you for bringing this to our attention. This sentence refers to the weaker mineral phases
that make up continental rocks versus the mafic oceanic crust. We have attempted to modify
this sentence, however, specifically listing mafic and felsic minerals reduces the quality of the
sentence.

Lines 92-93: In this sentence, it sounds like you are say that end-loading, basal traction, and
Insular superterrane collision 'all explicitly or implicitly favor a mechanically rigid continental
crust to transmit far-field stresses.' Is this what you mean to say?

Yes, all three original models favor a mechanically ridged crust to transmit far-field stresses. In
the discussion we present our modified basal traction model, involving strain decoupling in the
hot hinterland, and enhanced coupling of the Farallon slab with the colder, stronger foreland.

Line 94: no hyphen after 'Middle'

Thank you, we made this change.

Figures:

Figure 1: The Southern California Batholith is left off the Cordilleran arc in western North
America. It was a significant, 500 km-long belt of arc magma that was in 'flare-up' mode during
the beginning of the Laramide orogeny. I would add a pink field for the Mojave and Transverse

ranges and label it as the 'SC' or 'SCB' to denote that there was magmatism there in the Late
Cretaceous. This is of course an important part of the Laramide story...

Thank you, we included the Southern California Batholith.

Figure 4: The figure is fine, but you should be aware that the Farallon-Kula ridge was present
after 85 Ma. I'm not sure that you want to put it in this schematic figure, but depending on the
timing, the Farallon plate label may be on the 'Kula plate'. See Umhoefer (1987): Umhoefer,
P.J., 1987. Northward translation of "BAJA British Columbia" along the Late Cretaceous to
Paleocene margin of western North America. *Tectonics*, 6(4), pp.377-394.

You are correct, we originally did not want to over-interpret our schematic model. After your
suggestion we have included the approximate Farallon-Kula ridge based on Weil and Yonkee
(2023).

**Reviewer #2**

I have thoroughly reviewed the revised version of the manuscript "Hot Cordilleran hinterland
promoted lower crust mobility and decoupling of Laramide deformation", submitted by Vlaha et
al. Following my previous comments and suggestions, I have evaluated the authors' revisions
and the response letter that accompanied the submission.

I am pleased to report that the authors have addressed all the concerns raised during the initial
review. The revisions made to the manuscript, including adding extra discussion about the two
mechanisms, end-loading and basal shear, and the additional discussion regarding the other
reviewer's comments on the timing constraints on the Laramide events have significantly
improved the quality and clarity of the work. The authors have provided comprehensive
responses to the feedback, and the modifications made are well-justified and effectively
incorporated into the manuscript.

Based on the thoroughness of the revisions and the quality of the manuscript in its current form,
I recommend the manuscript for publication in *Nature communication*

This comment is greatly appreciated. Thank you for taking the time to review our manuscript.
